# Generalized Proximal Policy Optimization with Sample Reuse

**James Queeney**
Division of Systems Engineering
Boston University
`jqueeney@bu.edu`

**Ioannis Ch. Paschalidis**
Department of Electrical and Computer Engineering
Division of Systems Engineering
Boston University
`yannisp@bu.edu`

**Christos G. Cassandras**
Department of Electrical and Computer Engineering
Division of Systems Engineering
Boston University
`cgc@bu.edu`

## Abstract

In real-world decision making tasks, it is critical for data-driven reinforcement learning methods to be both stable and sample efficient. On-policy methods typically generate reliable policy improvement throughout training, while off-policy methods make more efficient use of data through sample reuse. In this work, we combine the theoretically supported stability benefits of on-policy algorithms with the sample efficiency of off-policy algorithms. We develop policy improvement guarantees that are suitable for the off-policy setting, and connect these bounds to the clipping mechanism used in Proximal Policy Optimization. This motivates an off-policy version of the popular algorithm that we call Generalized Proximal Policy Optimization with Sample Reuse. We demonstrate both theoretically and empirically that our algorithm delivers improved performance by effectively balancing the competing goals of stability and sample efficiency.

## 1   Introduction

In recent years, model-free deep reinforcement learning has been used to successfully solve complex simulated control tasks [4]. Unfortunately, real-world adoption of these techniques remains limited. High-stakes real-world decision making settings demand methods that deliver stable, reliable performance throughout training. In addition, real-world data collection can be difficult and expensive, so learning must make efficient use of limited data. The combination of these requirements is not an easy task, as stability and sample efficiency often represent competing interests. Existing model-free deep reinforcement learning algorithms often focus on one of these goals, and as a result sacrifice performance with respect to the other.

On-policy reinforcement learning methods such as Proximal Policy Optimization (PPO) [19] deliver stable performance throughout training due to their connection to theoretical policy improvement guarantees. These methods are motivated by a lower bound on the expected performance loss at every update, which can be approximated using samples generated by the current policy. The theoretically supported stability of these methods is very attractive, but the need for on-policy data and the high-variance nature of reinforcement learning often requires significant data to be collected between every update, resulting in high sample complexity and slow learning.

35th Conference on Neural Information Processing Systems (NeurIPS 2021).

Off-policy algorithms address the issue of high sample complexity by storing samples in a replay buffer, which allows data to be reused to calculate multiple policy updates. The ability to reuse samples improves learning speed, but also causes the distribution of data to shift away from the distribution generated by the current policy. This distribution shift invalidates the standard performance guarantees used in on-policy methods, and can lead to instability in the training process. Popular off-policy algorithms often require various implementation tricks and extensive hyperparameter tuning to control the instability caused by off-policy data.

By combining the attractive features of on-policy and off-policy methods in a principled way, we can balance the competing goals of stability and sample efficiency required in real-world decision making. We consider the popular on-policy algorithm PPO as our starting point due to its theoretically supported stable performance, and develop an off-policy variant with principled sample reuse that we call *Generalized Proximal Policy Optimization with Sample Reuse (GePPO)*. Our algorithm is based on the following main contributions:

1. We extend existing policy improvement guarantees to the off-policy setting, resulting in a lower bound that can be approximated using data from all recent policies.
2. We develop connections between the clipping mechanism used in PPO and the penalty term in our policy improvement lower bound, which motivates a generalized clipping mechanism for off-policy data.
3. We propose an adaptive learning rate method based on the same penalty term that more closely connects theory and practice.

We provide theoretical evidence that our algorithm effectively balances the goals of stability and sample efficiency, and we demonstrate the strong performance of our approach through experiments on high-dimensional continuous control tasks in OpenAI Gym's MuJoCo environments [3, 21].

## 2   Related work

**On-policy policy improvement methods**   The goal of monotonic policy improvement was first introduced by Kakade and Langford [14] in Conservative Policy Iteration, which maximizes a lower bound on policy improvement that can be constructed using samples from the current policy. This theory of policy improvement has served as a fundamental building block in the design of on-policy deep reinforcement learning methods, including the popular algorithms Trust Region Policy Optimization (TRPO) [17] and Proximal Policy Optimization (PPO) [19]. TRPO achieves approximate policy improvement by enforcing a Kullback-Leibler (KL) divergence trust region, while PPO does so by clipping the probability ratio between current and future policies.

Due to the strong performance of TRPO and PPO, there has been substantial interest in better understanding these methods. Engstrom et al. [5] and Andrychowicz et al. [2] both performed extensive empirical analysis on the various implementation choices in these algorithms, while other research has focused on the clipping mechanism used in PPO. Wang et al. [22] and Wang et al. [23] both proposed modifications to the clipping mechanism based on a KL divergence trust region. To the best of our knowledge, we are the first to directly relate the clipping mechanism in PPO to the total variation distance between policies. Wang et al. [23] also proposed a rollback operation to keep probability ratios close to the clipping range. We accomplish a similar goal by considering an adaptive learning rate.

**Sample efficiency with off-policy data**   A common approach to improving the sample efficiency of model-free reinforcement learning is to reuse samples collected under prior policies. Popular off-policy policy gradient approaches such as Deep Deterministic Policy Gradient (DDPG) [15], Twin Delayed DDPG (TD3) [8], and Soft Actor-Critic (SAC) [11] accomplish this by storing data in a replay buffer and sampling from this buffer to calculate policy updates. Note that these methods are not motivated by policy improvement guarantees, and do not directly control the bias introduced by off-policy data.

Other approaches have combined on-policy and off-policy policy gradients, with the goal of balancing the variance of on-policy methods and the bias of off-policy methods [7, 9, 10, 16, 24]. Gu et al. [10] demonstrated that the bias introduced by off-policy data is related to the KL divergence between the current policy and the behavior policy used to generate the data. Fakoor et al. [7] considered a related

KL divergence as a penalty term in their objective, while Wang et al. [24] approximately controlled this KL divergence by applying a trust region around a target policy. These methods are related to the penalty term that appears in our generalized policy improvement lower bound, which can be bounded by a penalty that depends on KL divergence.

Finally, there have been heuristic attempts to incorporate off-policy data into PPO [13, 20]. However, unlike our approach, these methods do not account for the distribution shift caused by off-policy data that invalidates the theoretical support for PPO.

## 3  Preliminaries

**Reinforcement learning framework**  We consider an infinite-horizon, discounted Markov Decision Process (MDP) defined by the tuple $(\mathcal{S}, \mathcal{A}, p, r, \rho_0, \gamma)$, where $\mathcal{S}$ is the set of states, $\mathcal{A}$ is the set of actions, $p : \mathcal{S} \times \mathcal{A} \to \Delta_{\mathcal{S}}$ is the transition probability function, $r : \mathcal{S} \times \mathcal{A} \to \mathbb{R}$ is the reward function, $\rho_0$ is the initial state distribution, and $\gamma$ is the discount rate.

We model the agent's decisions as a stationary policy $\pi : \mathcal{S} \to \Delta_{\mathcal{A}}$. Our goal is to choose a policy that maximizes the expected total discounted rewards $J(\pi) = \mathbb{E}_{\tau \sim \pi} \left[ \sum_{t=0}^{\infty} \gamma^t r(s_t, a_t) \right]$, where $\tau \sim \pi$ represents a trajectory sampled according to $s_0 \sim \rho_0$, $a_t \sim \pi(\cdot \mid s_t)$, and $s_{t+1} \sim p(\cdot \mid s_t, a_t)$. A policy $\pi$ also induces a normalized discounted state visitation distribution $d^\pi$, where $d^\pi(s) = (1 - \gamma) \sum_{t=0}^{\infty} \gamma^t \mathbb{P}(s_t = s \mid \rho_0, \pi, p)$. We write the corresponding normalized discounted state-action visitation distribution as $d^\pi(s, a) = d^\pi(s)\pi(a \mid s)$, where we make it clear from the context whether $d^\pi$ refers to a distribution over states or state-action pairs.

We denote the state value function of $\pi$ as $V^\pi(s) = \mathbb{E}_{\tau \sim \pi} \left[ \sum_{t=0}^{\infty} \gamma^t r(s_t, a_t) \mid s_0 = s \right]$, the state-action value function as $Q^\pi(s, a) = \mathbb{E}_{\tau \sim \pi} \left[ \sum_{t=0}^{\infty} \gamma^t r(s_t, a_t) \mid s_0 = s, a_0 = a \right]$, and the advantage function as $A^\pi(s, a) = Q^\pi(s, a) - V^\pi(s)$.

**Policy improvement lower bound**  The starting point in the design of many popular on-policy algorithms is the following policy improvement lower bound, which was first developed by Kakade and Langford [14] and later refined by Schulman et al. [17] and Achiam et al. [1]:

**Lemma 1** (Achiam et al. [1]).  *Consider a current policy $\pi_k$. For any future policy $\pi$, we have*

$$J(\pi) - J(\pi_k) \geq \frac{1}{1 - \gamma} \mathop{\mathbb{E}}_{(s,a) \sim d^{\pi_k}} \left[ \frac{\pi(a \mid s)}{\pi_k(a \mid s)} A^{\pi_k}(s, a) \right] - \frac{2\gamma C^{\pi, \pi_k}}{(1 - \gamma)^2} \mathop{\mathbb{E}}_{s \sim d^{\pi_k}} \left[ \mathrm{TV}(\pi, \pi_k)(s) \right], \quad (1)$$

*where $C^{\pi, \pi_k} = \max_{s \in \mathcal{S}} \left| \mathbb{E}_{a \sim \pi(\cdot \mid s)} \left[ A^{\pi_k}(s, a) \right] \right|$ and $\mathrm{TV}(\pi, \pi_k)(s)$ represents the total variation distance between the distributions $\pi(\cdot \mid s)$ and $\pi_k(\cdot \mid s)$.*

We refer to the first term of the lower bound in Lemma 1 as the surrogate objective, and the second term as the penalty term. Note that we can guarantee policy improvement at every step of the learning process by choosing the next policy $\pi_{k+1}$ to maximize this lower bound. Because the expectations in Lemma 1 depend on the current policy $\pi_k$, we can approximate this lower bound using samples generated by the current policy.

**Proximal Policy Optimization**  PPO, which has become the default on-policy policy optimization algorithm due to its strong performance and simple implementation, is theoretically motivated by the policy improvement lower bound in Lemma 1. Rather than directly maximizing this lower bound, PPO considers the goal of maximizing the surrogate objective while constraining the next policy to be close to the current policy. In particular, PPO heuristically accomplishes this by considering the following objective at every policy update:

$$L_k^{\mathrm{PPO}}(\pi) = \mathop{\mathbb{E}}_{(s,a) \sim d^{\pi_k}} \left[ \min \left( \frac{\pi(a \mid s)}{\pi_k(a \mid s)} A^{\pi_k}(s, a), \mathrm{clip} \left( \frac{\pi(a \mid s)}{\pi_k(a \mid s)}, 1 - \epsilon, 1 + \epsilon \right) A^{\pi_k}(s, a) \right) \right], \tag{2}$$

where $\mathrm{clip}(x, l, u) = \min(\max(x, l), u)$. As seen in the second term of this objective, PPO constrains the difference between consecutive policies by removing the incentive for the probability ratio $\pi(a|s)/\pi_k(a|s)$ to leave the clipping range $[1 - \epsilon, 1 + \epsilon]$. Finally, the outer minimization guarantees that (2) is a lower bound to the surrogate objective in Lemma 1. In practice, (2) is approximated using samples generated by the current policy $\pi_k$, and the resulting empirical objective is approximately optimized at every policy update using minibatch stochastic gradient ascent.

For a sufficiently small learning rate and sufficiently large number of samples, PPO results in stable policy improvement throughout the learning process. However, it is well-known that high variance is a major issue in reinforcement learning, so often the number of samples must be large in order for the empirical objective to be an accurate estimator of the true objective (2). Because these samples must be collected under the current policy between every policy update, PPO can be very sample intensive.

## 4 Generalized policy improvement lower bound

A logical approach to improve the sample efficiency of PPO is to reuse samples from prior policies, as done in off-policy algorithms. Unfortunately, the distribution shift between policies invalidates the policy improvement lower bound in Lemma 1, which provides the theoretical support for PPO's reliable performance. In order to retain the stability benefits of PPO while reusing samples from prior policies, we must incorporate these off-policy samples in a principled way. We accomplish this by developing a generalized policy improvement lower bound that can be approximated using samples from the last $M$ policies, rather than requiring samples be generated only from the current policy $\pi_k$:

**Theorem 1** (Generalized Policy Improvement Lower Bound). *Consider prior policies $\pi_{k-i}$, $i = 0, \ldots, M-1$, where $\pi_k$ represents the current policy. For any choice of distribution $\nu = [\nu_0 \cdots \nu_{M-1}]$ over the prior $M$ policies and any future policy $\pi$, we have*

$$
J(\pi) - J(\pi_k) \geq \frac{1}{1-\gamma} \mathop{\mathbb{E}}_{i \sim \nu} \left[ \mathop{\mathbb{E}}_{(s,a) \sim d^{\pi_{k-i}}} \left[ \frac{\pi(a \mid s)}{\pi_{k-i}(a \mid s)} A^{\pi_k}(s,a) \right] \right]
$$
$$
- \frac{2\gamma C^{\pi,\pi_k}}{(1-\gamma)^2} \mathop{\mathbb{E}}_{i \sim \nu} \left[ \mathop{\mathbb{E}}_{s \sim d^{\pi_{k-i}}} \left[ \mathrm{TV}(\pi, \pi_{k-i})(s) \right] \right], \quad (3)
$$

*where $C^{\pi,\pi_k}$ and $\mathrm{TV}(\pi, \pi_{k-i})(s)$ are defined as in Lemma 1.*

*Proof.* We generalize Lemma 1 to depend on expectations with respect to any reference policy, and we apply this result $M$ times where the reference policy is each of $\pi_{k-i}, i = 0, \ldots, M-1$, respectively. Then, the convex combination determined by $\nu$ of the resulting $M$ policy improvement lower bounds is also a lower bound. See the Appendix for a full proof. □

Because the expectations in Theorem 1 depend on distributions related to the last $M$ policies, this lower bound provides theoretical support for extending PPO to include off-policy samples. Note that Theorem 1 is still a lower bound on the policy improvement between the current policy $\pi_k$ and a future policy $\pi$. This is true because the advantage function in the surrogate objective and the constant in the penalty term still depend on the current policy $\pi_k$. However, the visitation distribution, probability ratio in the surrogate objective, and total variation distance in the penalty term now depend on prior policies. Finally, the standard policy improvement lower bound in Lemma 1 can be recovered from Theorem 1 by setting $M = 1$.

The penalty term in Theorem 1 suggests that we should control the expected total variation distances between the future policy $\pi$ and the last $M$ policies. By applying the triangle inequality for total variation distance to each component of the penalty term, we see that

$$
\mathop{\mathbb{E}}_{i \sim \nu} \left[ \mathop{\mathbb{E}}_{s \sim d^{\pi_{k-i}}} \left[ \mathrm{TV}(\pi, \pi_{k-i})(s) \right] \right] \leq \mathop{\mathbb{E}}_{i \sim \nu} \left[ \mathop{\mathbb{E}}_{s \sim d^{\pi_{k-i}}} \left[ \mathrm{TV}(\pi, \pi_k)(s) \right] \right]
$$
$$
+ \sum_{j=1}^{M-1} \sum_{i=j}^{M-1} \nu_i \mathop{\mathbb{E}}_{s \sim d^{\pi_{k-i}}} \left[ \mathrm{TV}(\pi_{k-j+1}, \pi_{k-j})(s) \right]. \quad (4)
$$

The first term on the right-hand side of (4) represents an expected total variation distance between the current policy $\pi_k$ and the future policy $\pi$, while each component of the second term represents an expected total variation distance between consecutive prior policies. This demonstrates that we can effectively control the expected performance loss at every policy update by controlling the expected total variation distance between consecutive policies. We see next that the clipping mechanism in PPO approximately accomplishes this task.

## 5 Clipping mechanism

**Connection to penalty term**   As discussed previously, the clipping mechanism present in the PPO objective removes the incentive for the probability ratio $\pi(a|s)/\pi_k(a|s)$ to leave the clipping range $[1-\epsilon, 1+\epsilon]$. Written differently, the clipping mechanism removes the incentive for the magnitude of

$$\left| \frac{\pi(a \mid s)}{\pi_k(a \mid s)} - 1 \right| \tag{5}$$

to exceed $\epsilon$. We now see that (5) is closely related to the penalty term of the standard policy improvement lower bound in Lemma 1 as follows:

**Lemma 2.** *The expected total variation distance between the current policy $\pi_k$ and the future policy $\pi$ that appears in Lemma 1 can be rewritten as*

$$\mathbb{E}_{s \sim d^{\pi_k}} \left[ \mathrm{TV}(\pi, \pi_k)(s) \right] = \frac{1}{2} \mathbb{E}_{(s,a) \sim d^{\pi_k}} \left[ \left| \frac{\pi(a \mid s)}{\pi_k(a \mid s)} - 1 \right| \right]. \tag{6}$$

*Proof.* See the Appendix. $\qquad\square$

Therefore, the clipping mechanism in PPO can be viewed as a heuristic that controls the magnitude of a sample-based approximation of the expectation on the right-hand side of Lemma 2. It accomplishes this by removing the incentive for (5) to exceed $\epsilon$ at all state-action pairs sampled from the state-action visitation distribution $d^{\pi_k}$. As a result, the clipping mechanism in PPO approximately bounds the expected total variation distance between the current policy $\pi_k$ and the future policy $\pi$ by $\epsilon/2$.

**Generalized clipping mechanism for off-policy data**   We can use this connection between the clipping mechanism and penalty term in PPO to derive a generalized clipping mechanism suitable for the off-policy setting. In particular, the decomposition of the off-policy penalty term from Theorem 1 that appears in (4) suggests that we should control the expected total variation distance

$$\mathbb{E}_{i \sim \nu} \left[ \mathbb{E}_{s \sim d^{\pi_{k-i}}} \left[ \mathrm{TV}(\pi, \pi_k)(s) \right] \right] \tag{7}$$

at each policy update, which can be rewritten as follows:

**Lemma 3.** *The expected total variation distance between the current policy $\pi_k$ and the future policy $\pi$ in (7) can be rewritten as*

$$\mathbb{E}_{i \sim \nu} \left[ \mathbb{E}_{s \sim d^{\pi_{k-i}}} \left[ \mathrm{TV}(\pi, \pi_k)(s) \right] \right] = \frac{1}{2} \mathbb{E}_{i \sim \nu} \left[ \mathbb{E}_{(s,a) \sim d^{\pi_{k-i}}} \left[ \left| \frac{\pi(a \mid s)}{\pi_{k-i}(a \mid s)} - \frac{\pi_k(a \mid s)}{\pi_{k-i}(a \mid s)} \right| \right] \right]. \tag{8}$$

*Proof.* Apply the same techniques as in the proof of Lemma 2. See the Appendix for details. $\qquad\square$

The right-hand side of Lemma 3 provides insight into the appropriate clipping mechanism to be applied in the off-policy setting in order to approximately control the penalty term in the generalized policy improvement lower bound:

**Definition 1** (Generalized Clipping Mechanism). *Consider a current policy $\pi_k$ and clipping parameter $\epsilon$. For a state-action pair generated using a prior policy $\pi_{k-i}$, the* generalized clipping mechanism *is defined as*

$$\mathrm{clip} \left( \frac{\pi(a \mid s)}{\pi_{k-i}(a \mid s)}, \frac{\pi_k(a \mid s)}{\pi_{k-i}(a \mid s)} - \epsilon, \frac{\pi_k(a \mid s)}{\pi_{k-i}(a \mid s)} + \epsilon \right). \tag{9}$$

Note that the probability ratio for each state-action pair in the off-policy setting begins in the center of the clipping range for every policy update just as in PPO, where now the center is given by $\pi_k(a|s)/\pi_{k-i}(a|s)$. Also note that we recover the standard clipping mechanism used in PPO when samples are generated by the current policy $\pi_k$.

**Impact of learning rate on clipping mechanism** Due to the heuristic nature of the clipping mechanism, it can only approximately bound the penalty term in the corresponding policy improvement lower bound if the learning rate used for policy updates is sufficiently small. To see why this is true, note that the clipping mechanism has no impact at the beginning of each policy update since each probability ratio begins at the center of the clipping range [5]. If the learning rate is too large, the initial gradient steps of the policy update can result in probability ratios that are far outside the clipping range. In addition, the sensitivity of the probability ratio to gradient updates can change as training progresses, which suggests that the learning rate may need to change over time in order for the clipping mechanism to approximately enforce a total variation distance trust region throughout the course of training.

In order to address these issues, we propose a simple adaptive learning rate method that is directly connected to our goal of controlling a total variation distance penalty term via the clipping mechanism. Using Lemma 3, we can approximate the expected total variation distance of interest using a sample-based estimate. We reduce the learning rate if the estimated total variation distance exceeds our goal of $\epsilon/2$, and we increase the learning rate if the estimated total variation distance is significantly lower than $\epsilon/2$. This approach more closely connects the implementation of PPO to the policy improvement lower bound on which the algorithm is based. In addition, the adaptive learning rate prevents large policy updates that can lead to instability, while also increasing the speed of learning when policy updates are too small. We formally describe this method in Algorithm 1.

## 6 Algorithm

The surrogate objective from the generalized policy improvement lower bound in Theorem 1, coupled with the generalized clipping mechanism in Definition 1 that controls the penalty term in Theorem 1, motivate the following generalized PPO objective that directly considers the use of off-policy data:

$$
L_k^{\text{GePPO}}(\pi) = \mathbb{E}_{i \sim \nu} \left[ \mathbb{E}_{(s,a) \sim d^{\pi_{k-i}}} \left[ \min \left( \frac{\pi(a \mid s)}{\pi_{k-i}(a \mid s)} A^{\pi_k}(s,a), \right. \right. \right.
$$
$$
\left. \left. \left. \text{clip} \left( \frac{\pi(a \mid s)}{\pi_{k-i}(a \mid s)}, \frac{\pi_k(a \mid s)}{\pi_{k-i}(a \mid s)} - \epsilon, \frac{\pi_k(a \mid s)}{\pi_{k-i}(a \mid s)} + \epsilon \right) A^{\pi_k}(s,a) \right) \right] \right]. \quad (10)
$$

Our algorithm, which we call Generalized Proximal Policy Optimization with Sample Reuse (GePPO), approximates this objective using samples collected from each of the last $M$ policies and approximately optimizes it using minibatch stochastic gradient ascent. In addition, we update the learning rate at every iteration using the adaptive method described in the previous section. GePPO, which we detail in Algorithm 1, represents a principled approach to improving the sample efficiency of PPO while retaining its approximate policy improvement guarantees.

## 7 Sample efficiency analysis

**Generalized clipping parameter** In order to compare GePPO to PPO, we first must determine the appropriate choice of clipping parameter $\epsilon^{\text{GePPO}}$ that results in the same worst-case expected performance loss at every update:

**Lemma 4.** *Consider PPO with clipping parameter $\epsilon^{\text{PPO}}$ and GePPO with clipping parameter $\epsilon^{\text{GePPO}}$. If*

$$
\epsilon^{\text{GePPO}} = \frac{\epsilon^{\text{PPO}}}{\mathbb{E}_{i \sim \nu}[i+1]}, \quad (11)
$$

*then the worst-case expected performance loss at every update is the same under both algorithms.*

*Proof.* See the Appendix. □

Lemma 4 shows that we must make smaller policy updates compared to PPO in terms of total variation distance, which is a result of utilizing samples from prior policies. However, these additional samples stabilize policy updates by increasing the batch size used to approximate the true objective, which allows us to make policy updates more frequently. Ultimately, this trade-off results in faster and more stable learning, as we detail next.

---

**Algorithm 1:** Generalized Proximal Policy Optimization with Sample Reuse (GePPO)

---

**Input:** initial policy $\pi_0$; number of prior policies $M$; policy weights $\nu$; clipping parameter $\epsilon$;
       batch size $n$; initial learning rate $\eta$; adaptive factor $\alpha \geq 0$; minimum threshold factor
       $0 \leq \beta \leq 1$.

**for** $k = 0, 1, 2, \ldots$ **do**

     Collect $n$ samples with $\pi_k$.

     Update policy:

     Use $n$ samples from each of $\pi_{k-i}$, $i = 0, \ldots, M - 1$, to approximate $L_k^{\text{GePPO}}(\pi)$.

     Approximately maximize the empirical objective using minibatch stochastic gradient ascent.

     Update learning rate:

     Calculate sample-based estimate $\widehat{\text{TV}}$ of expected total variation distance in Lemma 3.

     **if** $\widehat{\text{TV}} > \epsilon/2$ **then** $\eta = \eta \cdot 1/(1+\alpha)$ ;

     **else if** $\widehat{\text{TV}} < \beta\epsilon/2$ **then** $\eta = \eta \cdot (1 + \alpha)$.

**end**

---

**Balancing stability and sample efficiency**     For concreteness, in this section we restrict our attention to uniform policy weights over the last $M$ policies, i.e., $\nu_i = 1/M$ for $i = 0, \ldots, M - 1$. We provide additional details in the Appendix on how these policy weights can be optimized to further improve upon the results shown for the uniform case.

Assume we require $N = Bn$ samples for the empirical objective to be a sufficiently accurate estimate of the true objective, where $n$ is the smallest possible batch size we can collect and $B$ is some positive integer. In this setting, PPO makes one policy update per $N$ samples collected, while GePPO leverages data from prior policies to make $B$ updates per $N$ samples collected as long as $M \geq B$. First, we see that GePPO can increase the change in total variation distance of the policy throughout training compared to PPO without sacrificing stability in terms of sample size:

**Theorem 2.** *Set $M = B$ and consider uniform policy weights. Then, GePPO increases the change in total variation distance of the policy throughout training by a factor of $2B/(B+1)$ compared to PPO, while using the same number of samples for each policy update as PPO.*

*Proof.* See the Appendix.          $\square$

Alternatively, we see that GePPO can increase the sample size used to approximate each policy update compared to PPO, while maintaining the same change in total variation distance throughout training:

**Theorem 3.** *Set $M = 2B - 1$ and consider uniform policy weights. Then, GePPO increases the sample size used for each policy update by a factor of $(2B-1)/B$ compared to PPO, while maintaining the same change in total variation distance of the policy throughout training as PPO.*

*Proof.* See the Appendix.          $\square$

By combining the results in Theorem 2 and Theorem 3, we see that GePPO with uniform policy weights improves the trade-off between stability and sample efficiency in PPO for any choice of $B \leq M \leq 2B - 1$. Also note that the benefit of GePPO increases with $B$, where a larger value of $B$ indicates a more complex problem that requires additional samples to estimate the true objective at every policy update. This is precisely the scenario where the trade-off between stability and sample efficiency becomes critical.

## 8   Experiments

In addition to the theoretical support for our algorithm in the previous section, we aim to investigate the stability and sample efficiency of GePPO experimentally through simulations on several MuJoCo

Table 1: Performance comparison across MuJoCo tasks.

| Environment | Average Performance Over 1M Steps | | | Final Performance | | | Steps (M) to Final PPO Performance | |
|---|---|---|---|---|---|---|---|---|
| | PPO | GePPO | %* | PPO | GePPO | %* | PPO | GePPO |
| Swimmer-v3 | 98 | 161 | 65% | 136 | 195 | 44% | 1.00 | 0.23 |
| Hopper-v3 | 2,362 | 2,544 | 8% | 3,126 | 3,450 | 10% | 1.00 | 0.41 |
| HalfCheetah-v3 | 1,764 | 2,439 | 38% | 3,223 | 3,903 | 21% | 1.00 | 0.54 |
| Walker2d-v3 | 1,817 | 2,199 | 21% | 3,041 | 3,502 | 15% | 1.00 | 0.63 |
| Ant-v3 | 545 | 762 | 40% | 1,227 | 1,576 | 28% | 1.00 | 0.79 |
| Humanoid-v3 | 584 | 665 | 14% | 972 | 1,345 | 38% | 1.00 | 0.85 |

* Represents percent improvement of GePPO compared to PPO.

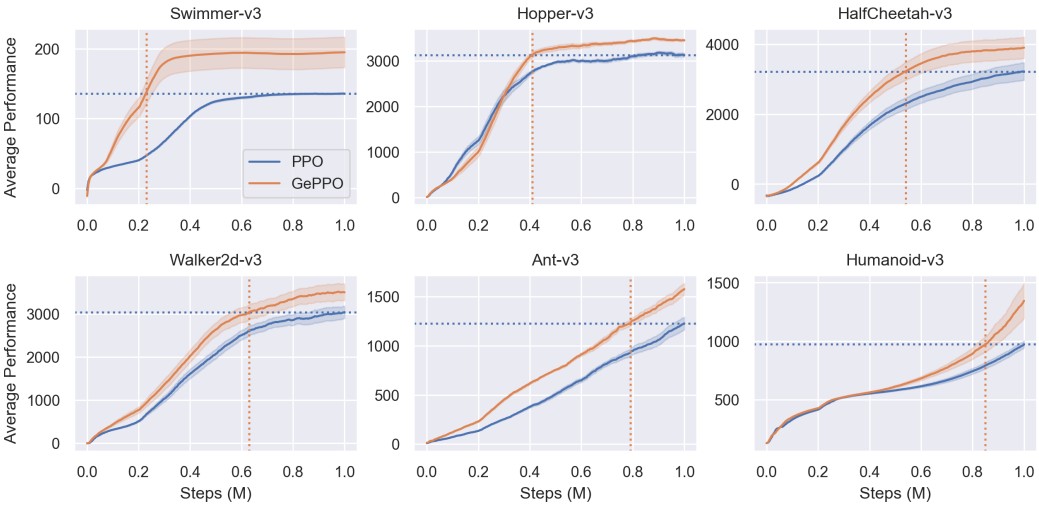

Figure 1: Performance throughout training across MuJoCo tasks. Shading denotes half of one standard error. Horizontal dotted lines represent the final performance of PPO, and vertical dotted lines represent the time at which GePPO achieves the same performance.

environments [21] in OpenAI Gym [3]. In particular, we consider six continuous control locomotion tasks which vary in dimensionality: Swimmer-v3, Hopper-v3, HalfCheetah-v3, Walker2d-v3, Ant-v3, and Humanoid-v3. We compare the performance of our algorithm to PPO, which is the default on-policy policy optimization algorithm. We do not consider a comparison with popular off-policy algorithms since they lack approximate policy improvement guarantees, and as a result the risk associated with each policy update is not comparable.

We consider the default implementation choices used by Henderson et al. [12] for PPO. In particular, we represent the policy $\pi$ as a multivariate Gaussian distribution, where the mean action for a given state is parameterized by a neural network with two hidden layers of 64 units each and tanh activations. The state-independent standard deviation is parameterized separately. The default value for the clipping parameter is $\epsilon^{\text{PPO}} = 0.2$, and the default batch size is $N = 2,048$. Sample trajectories for the tasks we consider can contain up to one thousand steps, so we represent the default batch size as $n = 1,024$ and $B = 2$ using the notation from the previous section.

For GePPO, we select $M$ and the corresponding policy weights $\nu$ to maximize the effective batch size used for policy updates while maintaining the same change in total variation distance throughout training as PPO. The clipping parameter $\epsilon^{\text{GePPO}}$ is chosen according to Lemma 4, which in our experiments results in $\epsilon^{\text{GePPO}} = 0.1$. We estimate $A^{\pi_k}(s, a)$ with an off-policy variant of Generalized Advantage Estimation [18] that uses the V-trace value function estimator [6]. We run each experiment

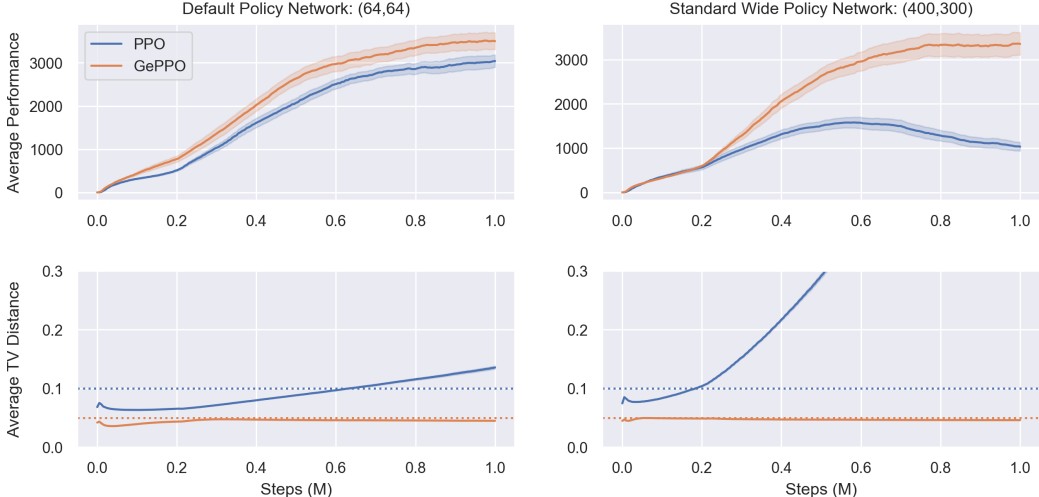

Figure 2: Evaluation on Walker2d-v3 with different policy networks. Hidden layer sizes in parentheses. Shading denotes half of one standard error. Top: Performance throughout training. Bottom: Change in average total variation distance per policy update. Horizontal dotted lines represent target total variation distances for PPO and GePPO, respectively.

for a total of one million steps over five random seeds. See the Appendix for additional implementation details, including the values of all hyperparameters.[1]

As shown in Table 1 and Figure 1, GePPO results in fast and reliable learning. We see that GePPO leads to improved final performance across all environments. In addition to final performance, we assess the sample efficiency of our algorithm by considering the average performance over the course of training as well as the number of samples required for GePPO to match the final performance of PPO. Despite the fact that the default implementation of PPO achieves stable learning with a small batch size ($B = 2$), our results still demonstrate the sample efficiency benefits of GePPO. Compared to PPO, GePPO improves the average performance over training by between 8% and 65%. Moreover, GePPO requires between 15% and 77% fewer samples to reach the final performance level of PPO.

In addition to improving sample efficiency, GePPO also ensures that the total variation distance between consecutive policies remains close to the target determined by the clipping parameter through the use of an adaptive learning rate. This is not the case in PPO, where we observe that the change in total variation distance per policy update increases throughout training. As shown on the left-hand side of Figure 2, the change in total variation distance for PPO under default settings is almost 40% higher than desired after one million steps on Walker2d-v3. We demonstrate that this can lead to instability in the training process by considering a standard wide policy network with two hidden layers of 400 and 300 units, respectively. We see on the right-hand side of Figure 2 that this minor implementation change exacerbates the trend observed in PPO under the default settings, resulting in unstable learning where performance declines over time due to excessively large policy updates. GePPO, on the other hand, successfully controls this source of instability through its adaptive learning rate, resulting in stable policy improvement that is robust to implementation choices.

# 9 Conclusion

We have presented a principled approach to incorporating off-policy samples into PPO that is theoretically supported by a novel off-policy policy improvement lower bound. Our algorithm, GePPO, improves the sample efficiency of PPO, and can be viewed as a more reliable approach to sample reuse than standard off-policy algorithms that are not based on approximate policy improvement guarantees. This represents an important step towards developing stable, sample efficient reinforcement learning methods that can be applied in high-stakes real-world decision making.

---

[1]Code available at https://github.com/jqueeney/geppo.

Despite this progress, there remain limitations that must be addressed in order for reinforcement learning to achieve widespread real-world adoption. Because our algorithm is based on PPO, policy improvement guarantees are only approximately achieved due to the use of the clipping mechanism heuristic. In addition, the constant factor in the penalty term on which our algorithm is based may be too large to deliver practical guarantees. Finally, we considered PPO as our starting point due to its theoretical support and stable performance, but there may exist other approaches that more effectively balance the goals of stability and sample efficiency. These represent interesting avenues for future work in order to develop reinforcement learning methods that can be trusted to improve real-world decision making.

## Acknowledgments and Disclosure of Funding

This research was partially supported by the NSF under grants ECCS-1931600, DMS-1664644, CNS-1645681, and IIS-1914792, by the ONR under grants N00014-19-1-2571 and N00014-21-1-2844, by the NIH under grants R01 GM135930 and UL54 TR004130, by the DOE under grants DE-AR-0001282 and NETL-EE0009696, by AFOSR under grant FA9550-19-1-0158, by the MathWorks, by the Boston University Kilachand Fund for Integrated Life Science and Engineering, and by the Qatar National Research Fund, a member of the Qatar Foundation (the statements made herein are solely the responsibility of the authors).

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
