# Appendix


## B  Proofs

### B.1  Proof of Theorem 1

We begin by stating results from Kakade and Langford [14] and Achiam et al. [1] that we will use in our proof.

**Lemma 5** (Kakade and Langford [14]). *Consider a current policy $\pi_k$. For any future policy $\pi$, we have*

$$J(\pi) - J(\pi_k) = \frac{1}{1-\gamma} \mathop{\mathbb{E}}_{s \sim d^\pi} \left[ \mathop{\mathbb{E}}_{a \sim \pi(\cdot|s)} \left[ A^{\pi_k}(s, a) \right] \right]. \tag{12}$$

**Lemma 6** (Achiam et al. [1]). *Consider a reference policy $\pi_{\text{ref}}$ and a future policy $\pi$. Then, the total variation distance between the state visitation distributions $d^{\pi_{\text{ref}}}$ and $d^{\pi}$ is bounded by*

$$\text{TV}(d^{\pi}, d^{\pi_{\text{ref}}}) \leq \frac{\gamma}{1-\gamma} \underset{s \sim d^{\pi_{\text{ref}}}}{\mathbb{E}} \left[ \text{TV}(\pi, \pi_{\text{ref}})(s) \right], \tag{13}$$

*where $\text{TV}(\pi, \pi_{\text{ref}})(s)$ is defined as in Lemma 1.*

Using these results, we first generalize Lemma 1 to depend on expectations with respect to any reference policy:

**Lemma 7.** *Consider a current policy $\pi_k$, and any reference policy $\pi_{\text{ref}}$. For any future policy $\pi$, we have*

$$J(\pi) - J(\pi_k) \geq \frac{1}{1-\gamma} \underset{(s,a) \sim d^{\pi_{\text{ref}}}}{\mathbb{E}} \left[ \frac{\pi(a \mid s)}{\pi_{\text{ref}}(a \mid s)} A^{\pi_k}(s, a) \right] - \frac{2\gamma C^{\pi, \pi_k}}{(1-\gamma)^2} \underset{s \sim d^{\pi_{\text{ref}}}}{\mathbb{E}} \left[ \text{TV}(\pi, \pi_{\text{ref}})(s) \right], \tag{14}$$

*where $C^{\pi, \pi_k}$ and $\text{TV}(\pi, \pi_{\text{ref}})(s)$ are defined as in Lemma 1.*

*Proof.* The proof is similar to the proof of Lemma 1 in Achiam et al. [1]. Starting from the equality in Lemma 5, we add and subtract the term

$$\frac{1}{1-\gamma} \underset{s \sim d^{\pi_{\text{ref}}}}{\mathbb{E}} \left[ \underset{a \sim \pi(\cdot \mid s)}{\mathbb{E}} \left[ A^{\pi_k}(s, a) \right] \right]. \tag{15}$$

By doing so, we have

$$\begin{aligned}
J(\pi) - J(\pi_k) &= \frac{1}{1-\gamma} \underset{s \sim d^{\pi_{\text{ref}}}}{\mathbb{E}} \left[ \underset{a \sim \pi(\cdot \mid s)}{\mathbb{E}} \left[ A^{\pi_k}(s, a) \right] \right] \\
&\quad + \frac{1}{1-\gamma} \left( \underset{s \sim d^{\pi}}{\mathbb{E}} \left[ \underset{a \sim \pi(\cdot \mid s)}{\mathbb{E}} \left[ A^{\pi_k}(s, a) \right] \right] - \underset{s \sim d^{\pi_{\text{ref}}}}{\mathbb{E}} \left[ \underset{a \sim \pi(\cdot \mid s)}{\mathbb{E}} \left[ A^{\pi_k}(s, a) \right] \right] \right) \\
&\geq \frac{1}{1-\gamma} \underset{s \sim d^{\pi_{\text{ref}}}}{\mathbb{E}} \left[ \underset{a \sim \pi(\cdot \mid s)}{\mathbb{E}} \left[ A^{\pi_k}(s, a) \right] \right] \\
&\quad - \frac{1}{1-\gamma} \left| \underset{s \sim d^{\pi}}{\mathbb{E}} \left[ \underset{a \sim \pi(\cdot \mid s)}{\mathbb{E}} \left[ A^{\pi_k}(s, a) \right] \right] - \underset{s \sim d^{\pi_{\text{ref}}}}{\mathbb{E}} \left[ \underset{a \sim \pi(\cdot \mid s)}{\mathbb{E}} \left[ A^{\pi_k}(s, a) \right] \right] \right|.
\end{aligned} \tag{16}$$

We can bound the second term in (16) using Hölder's inequality:

$$\begin{aligned}
\frac{1}{1-\gamma} &\left| \underset{s \sim d^{\pi}}{\mathbb{E}} \left[ \underset{a \sim \pi(\cdot \mid s)}{\mathbb{E}} \left[ A^{\pi_k}(s, a) \right] \right] - \underset{s \sim d^{\pi_{\text{ref}}}}{\mathbb{E}} \left[ \underset{a \sim \pi(\cdot \mid s)}{\mathbb{E}} \left[ A^{\pi_k}(s, a) \right] \right] \right| \\
&\leq \frac{1}{1-\gamma} \| d^{\pi} - d^{\pi_{\text{ref}}} \|_1 \left\| \underset{a \sim \pi(\cdot \mid s)}{\mathbb{E}} \left[ A^{\pi_k}(s, a) \right] \right\|_{\infty}, \tag{17}
\end{aligned}$$

where $d^{\pi}$ and $d^{\pi_{\text{ref}}}$ represent state visitation distributions. From the definition of total variation distance and Lemma 6, we have

$$\| d^{\pi} - d^{\pi_{\text{ref}}} \|_1 = 2 \, \text{TV}(d^{\pi}, d^{\pi_{\text{ref}}}) \leq \frac{2\gamma}{1-\gamma} \underset{s \sim d^{\pi_{\text{ref}}}}{\mathbb{E}} \left[ \text{TV}(\pi, \pi_{\text{ref}})(s) \right]. \tag{18}$$

Also note that

$$\left\| \underset{a \sim \pi(\cdot \mid s)}{\mathbb{E}} \left[ A^{\pi_k}(s, a) \right] \right\|_{\infty} = \max_{s \in \mathcal{S}} \left| \underset{a \sim \pi(\cdot \mid s)}{\mathbb{E}} \left[ A^{\pi_k}(s, a) \right] \right| = C^{\pi, \pi_k}. \tag{19}$$

As a result, we have that

$$J(\pi) - J(\pi_k) \geq \frac{1}{1-\gamma} \underset{s \sim d^{\pi_{\text{ref}}}}{\mathbb{E}} \left[ \underset{a \sim \pi(\cdot \mid s)}{\mathbb{E}} \left[ A^{\pi_k}(s, a) \right] \right] - \frac{2\gamma C^{\pi, \pi_k}}{(1-\gamma)^2} \underset{s \sim d^{\pi_{\text{ref}}}}{\mathbb{E}} \left[ \text{TV}(\pi, \pi_{\text{ref}})(s) \right]. \tag{20}$$

Finally, assume that the support of $\pi$ is contained in the support of $\pi_{\text{ref}}$ for all states, which is true for common policy representations used in policy optimization. Then, we can rewrite the first term on the right-hand side of (20) as

$$\frac{1}{1-\gamma} \underset{s \sim d^{\pi_{\text{ref}}}}{\mathbb{E}} \left[ \underset{a \sim \pi(\cdot \mid s)}{\mathbb{E}} \left[ A^{\pi_k}(s, a) \right] \right] = \frac{1}{1-\gamma} \underset{(s,a) \sim d^{\pi_{\text{ref}}}}{\mathbb{E}} \left[ \frac{\pi(a \mid s)}{\pi_{\text{ref}}(a \mid s)} A^{\pi_k}(s, a) \right], \tag{21}$$

which results in the lower bound in (14). □

We are now ready to prove Theorem 1:

*Proof of Theorem 1.* Consider prior policies $\pi_{k-i}$, $i = 0, \ldots, M-1$. For each prior policy, assume that the support of $\pi$ is contained in the support of $\pi_{k-i}$ for all states, which is true for common policy representations used in policy optimization. Then, by Lemma 7 we have

$$J(\pi) - J(\pi_k) \geq \frac{1}{1-\gamma} \mathop{\mathbb{E}}_{(s,a) \sim d^{\pi_{k-i}}} \left[ \frac{\pi(a \mid s)}{\pi_{k-i}(a \mid s)} A^{\pi_k}(s,a) \right] - \frac{2\gamma C^{\pi,\pi_k}}{(1-\gamma)^2} \mathop{\mathbb{E}}_{s \sim d^{\pi_{k-i}}} \left[ \mathrm{TV}(\pi, \pi_{k-i})(s) \right] \tag{22}$$

for each $\pi_{k-i}$, $i = 0, \ldots, M-1$. Consider policy weights $\nu = [\nu_0 \cdots \nu_{M-1}]$ over the last $M$ policies, where $\nu$ is a distribution. Then, for any choice of distribution $\nu$, the convex combination determined by $\nu$ of the $M$ lower bounds given by (22) results in the lower bound

$$J(\pi) - J(\pi_k) \geq \frac{1}{1-\gamma} \mathop{\mathbb{E}}_{i \sim \nu} \left[ \mathop{\mathbb{E}}_{(s,a) \sim d^{\pi_{k-i}}} \left[ \frac{\pi(a \mid s)}{\pi_{k-i}(a \mid s)} A^{\pi_k}(s,a) \right] \right]$$
$$- \frac{2\gamma C^{\pi,\pi_k}}{(1-\gamma)^2} \mathop{\mathbb{E}}_{i \sim \nu} \left[ \mathop{\mathbb{E}}_{s \sim d^{\pi_{k-i}}} \left[ \mathrm{TV}(\pi, \pi_{k-i})(s) \right] \right]. \tag{23}$$

$\square$

## B.2 Proof of Lemma 2

*Proof.* From the definition of total variation distance, we have that

$$\mathop{\mathbb{E}}_{s \sim d^{\pi_k}} \left[ \mathrm{TV}(\pi, \pi_k)(s) \right] = \mathop{\mathbb{E}}_{s \sim d^{\pi_k}} \left[ \frac{1}{2} \int_{a \in \mathcal{A}} |\pi(a \mid s) - \pi_k(a \mid s)| \, \mathrm{d}a \right]. \tag{24}$$

Assume that the support of $\pi$ is contained in the support of $\pi_k$ for all states, which is true for common policy representations used in policy optimization. Then, by multiplying and dividing by $\pi_k(a \mid s)$, we see that

$$\mathop{\mathbb{E}}_{s \sim d^{\pi_k}} \left[ \mathrm{TV}(\pi, \pi_k)(s) \right] = \mathop{\mathbb{E}}_{s \sim d^{\pi_k}} \left[ \frac{1}{2} \int_{a \in \mathcal{A}} \pi_k(a \mid s) \left| \frac{\pi(a \mid s)}{\pi_k(a \mid s)} - 1 \right| \mathrm{d}a \right]$$
$$= \frac{1}{2} \mathop{\mathbb{E}}_{(s,a) \sim d^{\pi_k}} \left[ \left| \frac{\pi(a \mid s)}{\pi_k(a \mid s)} - 1 \right| \right]. \tag{25}$$

$\square$

## B.3 Proof of Lemma 3

*Proof.* From the definition of total variation distance, we have that

$$\mathop{\mathbb{E}}_{i \sim \nu} \left[ \mathop{\mathbb{E}}_{s \sim d^{\pi_{k-i}}} \left[ \mathrm{TV}(\pi, \pi_k)(s) \right] \right] = \mathop{\mathbb{E}}_{i \sim \nu} \left[ \mathop{\mathbb{E}}_{s \sim d^{\pi_{k-i}}} \left[ \frac{1}{2} \int_{a \in \mathcal{A}} |\pi(a \mid s) - \pi_k(a \mid s)| \, \mathrm{d}a \right] \right]. \tag{26}$$

For $i = 0, \ldots, M-1$, assume that the supports of $\pi$ and $\pi_k$ are contained in the support of $\pi_{k-i}$ for all states, which is true for common policy representations used in policy optimization. Then, by multiplying and dividing by $\pi_{k-i}(a \mid s)$, we see that

$$\mathop{\mathbb{E}}_{i \sim \nu} \left[ \mathop{\mathbb{E}}_{s \sim d^{\pi_{k-i}}} \left[ \mathrm{TV}(\pi, \pi_k)(s) \right] \right]$$
$$= \mathop{\mathbb{E}}_{i \sim \nu} \left[ \mathop{\mathbb{E}}_{s \sim d^{\pi_{k-i}}} \left[ \frac{1}{2} \int_{a \in \mathcal{A}} \pi_{k-i}(a \mid s) \left| \frac{\pi(a \mid s)}{\pi_{k-i}(a \mid s)} - \frac{\pi_k(a \mid s)}{\pi_{k-i}(a \mid s)} \right| \mathrm{d}a \right] \right] \tag{27}$$
$$= \frac{1}{2} \mathop{\mathbb{E}}_{i \sim \nu} \left[ \mathop{\mathbb{E}}_{(s,a) \sim d^{\pi_{k-i}}} \left[ \left| \frac{\pi(a \mid s)}{\pi_{k-i}(a \mid s)} - \frac{\pi_k(a \mid s)}{\pi_{k-i}(a \mid s)} \right| \right] \right].$$

$\square$

### B.4 Proof of Lemma 4

*Proof.* From Lemma 2, PPO approximately bounds the penalty term in the standard policy improvement lower bound in Lemma 1 by

$$\frac{2\gamma C^{\pi,\pi_k}}{(1-\gamma)^2} \mathop{\mathbb{E}}_{s\sim d^{\pi_k}} \left[ \mathrm{TV}(\pi,\pi_k)(s) \right] \leq \frac{2\gamma C^{\pi,\pi_k}}{(1-\gamma)^2} \cdot \frac{\epsilon^{\mathrm{PPO}}}{2}. \tag{28}$$

Using the triangle inequality for total variation distance, we see that the penalty term in the generalized policy improvement lower bound in Theorem 1 can be bounded by

$$\frac{2\gamma C^{\pi,\pi_k}}{(1-\gamma)^2} \mathop{\mathbb{E}}_{i\sim\nu} \left[ \mathop{\mathbb{E}}_{s\sim d^{\pi_{k-i}}} \left[ \mathrm{TV}(\pi,\pi_{k-i})(s) \right] \right]$$
$$\leq \frac{2\gamma C^{\pi,\pi_k}}{(1-\gamma)^2} \mathop{\mathbb{E}}_{i\sim\nu} \left[ \sum_{j=0}^{i} \mathop{\mathbb{E}}_{s\sim d^{\pi_{k-i}}} \left[ \mathrm{TV}(\pi_{k-j+1},\pi_{k-j})(s) \right] \right], \tag{29}$$

where we have written the future policy $\pi$ as $\pi_{k+1}$ on the right-hand side. Note that policy updates in GePPO approximately bound each expected total variation distance that appears on the right-hand side of (29) by $\epsilon^{\mathrm{GePPO}}/2$. Therefore, the penalty term in the generalized policy improvement lower bound is approximately bounded by

$$\frac{2\gamma C^{\pi,\pi_k}}{(1-\gamma)^2} \mathop{\mathbb{E}}_{i\sim\nu} \left[ \mathop{\mathbb{E}}_{s\sim d^{\pi_{k-i}}} \left[ \mathrm{TV}(\pi,\pi_{k-i})(s) \right] \right] \leq \frac{2\gamma C^{\pi,\pi_k}}{(1-\gamma)^2} \mathop{\mathbb{E}}_{i\sim\nu} \left[ \frac{\epsilon^{\mathrm{GePPO}}}{2} \cdot (i+1) \right]$$
$$= \frac{2\gamma C^{\pi,\pi_k}}{(1-\gamma)^2} \cdot \frac{\epsilon^{\mathrm{GePPO}}}{2} \cdot \mathop{\mathbb{E}}_{i\sim\nu} \left[ i+1 \right]. \tag{30}$$

By comparing the bounds in (28) and (30), we see that the worst-case expected performance loss at every update is the same for PPO and GePPO when

$$\epsilon^{\mathrm{GePPO}} = \frac{\epsilon^{\mathrm{PPO}}}{\mathbb{E}_{i\sim\nu} \left[ i+1 \right]}. \tag{31}$$

$\square$

### B.5 Proof of Theorem 2

*Proof.* For $M = B$ with uniform policy weights, we see by Lemma 4 that

$$\epsilon^{\mathrm{GePPO}} = \frac{\epsilon^{\mathrm{PPO}}}{\frac{1}{B} \sum_{i=0}^{B-1} (i+1)} = \frac{2}{B+1} \cdot \epsilon^{\mathrm{PPO}}. \tag{32}$$

PPO makes one policy update per $N = Bn$ samples collected, which results in a policy change of $\epsilon^{\mathrm{PPO}}/2$ in terms of total variation distance. By leveraging data from prior policies to obtain $N$ samples per update, GePPO makes $B$ policy updates per $N$ samples collected. This results in an overall policy change of

$$B \cdot \frac{\epsilon^{\mathrm{GePPO}}}{2} = \frac{2B}{B+1} \cdot \frac{\epsilon^{\mathrm{PPO}}}{2} \tag{33}$$

in terms of total variation distance for every $N$ samples collected. Therefore, GePPO increases the change in total variation distance of the policy throughout training by a factor of $2B/(B+1)$ compared to PPO, while using the same number of samples for each policy update. $\square$

### B.6 Proof of Theorem 3

*Proof.* Because $M = 2B - 1$, GePPO uses $(2B - 1)n$ samples to compute each policy update, compared to $N = Bn$ samples used in PPO. Therefore, GePPO increases the sample size used for each policy update by a factor of $(2B-1)/B$ compared to PPO.

For $M = 2B - 1$ with uniform policy weights, we see by Lemma 4 that

$$\epsilon^{\mathrm{GePPO}} = \frac{\epsilon^{\mathrm{PPO}}}{\frac{1}{2B-1} \sum_{i=0}^{2B-2} (i+1)} = \frac{\epsilon^{\mathrm{PPO}}}{B}. \tag{34}$$

As in Theorem 2, PPO makes one policy update per $N$ samples collected, while GePPO makes $B$ policy updates per $N$ samples collected. This results in an overall change in total variation distance of

$$B \cdot \frac{\epsilon^{\text{GePPO}}}{2} = \frac{\epsilon^{\text{PPO}}}{2}, \tag{35}$$

which is the same as in PPO. $\qquad\square$

## C  Optimal policy weights

In Section 7, we demonstrated the benefits of our algorithm with uniform policy weights, i.e., $\nu_i = {}^1\!/\!_M$ for $i = 0, \dots, M-1$. Because our generalized policy improvement lower bound holds for any choice of policy weights $\nu$, we can improve upon the results in Theorem 2 and Theorem 3 by optimizing $\nu$.

Non-uniform policy weights introduce an additional source of variance, so in order to account for this we must extend the notion of sample size to effective sample size. The effective sample size represents the number of uniformly-weighted samples that result in the same level of variance. For $n$ samples collected under each of the prior $M$ policies, the effective sample size used in the empirical objective of GePPO with policy weights $\nu$ can be written as

$$\text{ESS}^{\text{GePPO}} = \frac{n}{\sum_{i=0}^{M-1} \nu_i^2}. \tag{36}$$

Note that the effective sample size of GePPO with non-uniform policy weights is always smaller than the true number of samples used to calculate the empirical objective, i.e., $\text{ESS}^{\text{GePPO}} < Mn$ unless $\nu$ is the uniform distribution.

By Lemma 4, GePPO results in an overall policy change of

$$B \cdot \frac{\epsilon^{\text{GePPO}}}{2} = \frac{B}{\sum_{i=0}^{M-1} \nu_i(i+1)} \cdot \frac{\epsilon^{\text{PPO}}}{2} \tag{37}$$

in terms of total variation distance for every $N = Bn$ samples collected, as long as the effective sample size of GePPO is at least $N$. Using the general forms of effective sample size in (36) and total variation distance change in (37), we can optimize the results in Theorem 2 as follows:

**Theorem 4.** *Let $\bar{M} \geq B$ be the maximum number of prior policies. Consider the goal of maximizing the change in total variation distance of the policy with GePPO while maintaining the same effective sample size of $N = Bn$ used in PPO. Then, the policy weights $\nu$ that achieve this goal are the optimal solution to the convex optimization problem*

$$\min_{\nu_0, \dots, \nu_{\bar{M}-1}} \sum_{i=0}^{\bar{M}-1} \nu_i(i+1)$$
$$\text{s.t.} \quad \sum_{i=0}^{\bar{M}-1} \nu_i^2 \leq \frac{1}{B}, \quad \sum_{i=0}^{\bar{M}-1} \nu_i = 1, \tag{38}$$
$$\nu_i \geq 0, \ i = 0, \dots, \bar{M}-1.$$

*Proof.* We can maximize the total variation distance change in (37) by choosing $\nu$ that minimizes the denominator, leading to the objective in (38). Next, we must have an effective sample size that is at least $N = Bn$. Equivalently, we need

$$\text{ESS}^{\text{GePPO}} = \frac{n}{\sum_{i=0}^{\bar{M}-1} \nu_i^2} \geq Bn, \tag{39}$$

which can be rewritten as the first constraint in (38). Finally, the other constraints in (38) ensure that $\nu$ is a distribution. $\qquad\square$

Theorem 2 considers uniform policy weights with $M = B$, which are a feasible solution to (38). As a result, Theorem 4 increases the change in total variation distance compared to Theorem 2.

Similarly, we can optimize the results in Theorem 3 as follows:

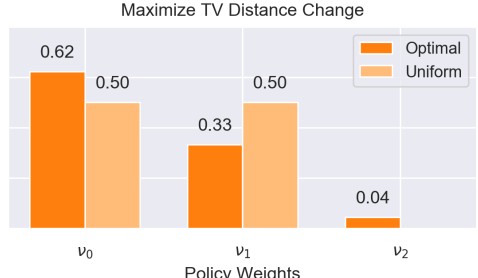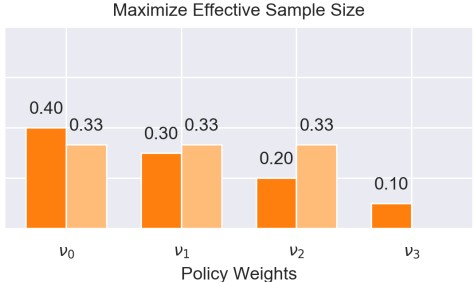

Figure 3: Comparison of optimal and uniform policy weights for GePPO when $B = 2$. Left: Policy weights determined by Theorem 4 and Theorem 2, respectively. Right: Policy weights determined by Theorem 5 and Theorem 3, respectively.

**Theorem 5.** *Let $\bar{M} \geq 2B - 1$ be the maximum number of prior policies. Consider the goal of maximizing the effective sample size with GePPO while maintaining the same change in total variation distance of the policy throughout training as PPO. Then, the policy weights $\nu$ that achieve this goal are the optimal solution to the convex optimization problem*

$$\min_{\nu_0,\ldots,\nu_{\bar{M}-1}} \sum_{i=0}^{\bar{M}-1} \nu_i^2$$
$$\text{s.t.} \quad \sum_{i=0}^{\bar{M}-1} \nu_i(i+1) = B, \quad \sum_{i=0}^{\bar{M}-1} \nu_i = 1, \tag{40}$$
$$\nu_i \geq 0, \ i = 0,\ldots,\bar{M}-1.$$

*Proof.* We can maximize the effective sample size in (36) by choosing $\nu$ that minimizes the denominator, leading to the objective in (40). Next, the total variation distance change in (37) must equal $\epsilon^{\text{PPO}}/2$, which we accomplish with the first constraint in (40). Finally, the other constraints in (40) ensure that $\nu$ is a distribution. □

Theorem 3 considers uniform policy weights with $M = 2B - 1$, which are a feasible solution to (40). Therefore, Theorem 5 increases the effective sample size compared to Theorem 3. Also note that by selecting $\bar{M}$ to be large, Theorem 4 and Theorem 5 solve for both the optimal choice of $M$ and the corresponding optimal weights.

See Figure 3 for a comparison of the optimal and uniform policy weights when $B = 2$, which is the setting for the experiments we considered. Note that the optimal policy weights can be found very efficiently, and the convex optimization problems in Theorem 4 and Theorem 5 only need to be solved once at the beginning of training to determine the policy weights $\nu$.

## D  Implementation details

To aid in reproducibility, we describe the implementation details used to produce our experimental results. Note that all choices are based on the default implementation of PPO in Henderson et al. [12].

**Network structures and hyperparameters**  As discussed in Section 8, we represent the policy $\pi$ as a multivariate Gaussian distribution where the mean action for a given state is parameterized by a neural network with two hidden layers of 64 units each and tanh activations. The state-independent standard deviation is parameterized separately, where for each action dimension the standard deviation is initialized as a multiple of half of the feasible action range. The value function $V^\pi(s)$ is parameterized by a separate neural network with two hidden layers of 64 units each and tanh activations. Observations are standardized using a running mean and standard deviation throughout the training process. Both the policy and value function are updated at every iteration using minibatch stochastic gradient descent. All hyperparameters associated with these optimization processes can

Table 2: Hyperparameter values for experimental results.

| General | Default | Ant | Humanoid |
|---|---|---|---|
| Discount rate ($\gamma$) | 0.995 | | |
| GAE parameter ($\lambda$) | 0.97 | | |
| Minibatches per epoch | 32 | | |
| Epochs per update | 10 | | |
| Value function optimizer | Adam | | |
| Value function learning rate | 3e−4 | | |
| Policy optimizer | Adam | | |
| Initial policy learning rate ($\eta$) | 3e−4 | 1e−4 | 3e−5 |
| Initial policy std. deviation multiple | 1.0 | 0.5 | 0.5 |

| PPO | | | |
|---|---|---|---|
| Clipping parameter ($\epsilon^{\text{PPO}}$) | 0.2 | | |
| Batch size ($N$) | 2,048 | | |

| GePPO | | | |
|---|---|---|---|
| Clipping parameter ($\epsilon^{\text{GePPO}}$) * | 0.1 | | |
| Number of prior policies ($M$) * | 4 | | |
| Minimum batch size ($n$) | 1,024 | | |
| Adaptive factor ($\alpha$) | 0.03 | | |
| Minimum threshold factor ($\beta$) | 0.5 | | |
| V-trace truncation parameter ($\bar{c}$) | 1.0 | | |

\* Represents calculated value.

be found in Table 2. Due to the high-dimensional nature of Ant-v3 and Humanoid-v3, we tuned the initial learning rate and standard deviation multiple of the policy used for PPO on these tasks. These values can also be found in Table 2. For a fair comparison, we used the same hyperparameter values for GePPO.

Following Henderson et al. [12], we consider the clipping parameter $\epsilon^{\text{PPO}} = 0.2$ and a batch size of $N = 2{,}048$ for PPO. Because sample trajectories for the tasks we consider can contain up to one thousand steps, we consider a minimum batch size of $n = 1{,}024$ for GePPO. When writing the default batch size in PPO as $N = Bn$, this results in $B = 2$. Using this value of $B$, the number of prior policies $M$ and the corresponding policy weights $\nu$ for GePPO are calculated according to Theorem 5. These weights are shown on the right-hand side of Figure 3. The clipping parameter $\epsilon^{\text{GePPO}}$ is calculated based on these policy weights using Lemma 4, which results in $\epsilon^{\text{GePPO}} = 0.1$. Finally, the adaptive factor $\alpha$ and minimum threshold factor $\beta$ used for our adaptive learning rate method can be found in Table 2.

**Advantage estimation**   Advantages $A^{\pi_k}(s, a)$ of the current policy in PPO are estimated using Generalized Advantage Estimation (GAE) [18] with $\lambda = 0.97$. Note that the parameter $\lambda$ in GAE determines a weighted average over $K$-step advantage estimates. When samples are collected using the current policy $\pi_k$, these multi-step advantage estimates are unbiased except for the use of bootstrapping with the learned value function. In GePPO, however, we must estimate $A^{\pi_k}(s, a)$ using samples collected from prior policies, so the multi-step advantage estimates used in GAE are no longer unbiased. Instead, we use V-trace [6] to calculate corrected estimates that are suitable for the off-policy setting. V-trace corrects multi-step estimates while controlling variance by using truncated importance sampling. For a learned value function $V$, current policy $\pi_k$, and prior policy $\pi_{k-i}$ used to generate the data, this results in the following $K$-step target for the value function:

$$V_{\text{trace}}^{\pi_k}(s_t) = V(s_t) + \sum_{j=0}^{K-1} \gamma^j \left( \prod_{i=0}^{j} c_{t+i} \right) \delta_{t+j}^{V}, \tag{41}$$

where $\delta_t^V = r(s_t, a_t) + \gamma V(s_{t+1}) - V(s_t)$ and $c_t = \min\left(\bar{c}, {\pi_k(a_t|s_t)}/{\pi_{k-i}(a_t|s_t)}\right)$ represents a truncated importance sampling ratio with truncation parameter $\bar{c}$. We can use the same correction techniques to generate $K$-step advantage estimates from off-policy data. For $K \geq 2$, the corresponding $K$-step advantage estimate is given by

$$A_{\text{trace}}^{\pi_k}(s_t, a_t) = \delta_t^V + \sum_{j=1}^{K-1} \gamma^j \left(\prod_{i=1}^{j} c_{t+i}\right) \delta_{t+j}^V, \tag{42}$$

and for $K = 1$ we have the standard one-step estimate $A_{\text{trace}}^{\pi_k}(s_t, a_t) = \delta_t^V$ that does not require any correction. We use $\bar{c} = 1.0$ in our experiments, which is the default setting in Espeholt et al. [6]. Note that Espeholt et al. [6] treat the final importance sampling ratio in each term separately, but in practice the truncation parameters are chosen to be the same so we do not make this distinction in our notation. Finally, we consider a weighted average over these corrected multi-step advantage estimates as in GAE.

Typically, the resulting advantage estimates are standardized within each minibatch of PPO. Note that the expectation of $A^{\pi_k}(s, a)$ with respect to samples generated by $\pi_k$ is zero, so the centering of advantage estimates ensures that the empirical average also satisfies this property. In the off-policy setting, the appropriate quantity to standardize is the starting point of policy updates

$$\frac{\pi_k(a \mid s)}{\pi_{k-i}(a \mid s)} A^{\pi_k}(s, a), \tag{43}$$

since the expectation of (43) with respect to samples generated by prior policies is zero. Note that the standardization of (43) generalizes the standardization done in PPO, where the probability ratios at the beginning of each policy update are all equal to one.

**Computational resources**   All experiments were run on a Windows 10 operating system, Intel Core i7-9700 CPU with base speed of 3.0 GHz and 32 GB of RAM, and NVIDIA GeForce RTX 2060 GPU with 6 GB of dedicated memory. OpenAI Gym [3] is available under The MIT License, and we make use of MuJoCo [21] with a license obtained from Roboti LLC. Using code that has not been optimized for execution speed, simulations for all seeds on a given environment required approximately 3 hours of wall-clock time for PPO and 4 hours of wall-clock time for GePPO. The increased wall-clock time of GePPO is due to the fact that GePPO performs twice as many updates in our experiments compared to PPO. Note that on average the sample efficiency benefits of GePPO not only lead to improved performance compared to PPO for a fixed number of samples, but also for a fixed amount of wall-clock time.

## E   Additional experimental results

In this section, we provide additional experimental results to further analyze the performance of our algorithm. In particular, we include results across all MuJoCo tasks for both the default policy network with two hidden layers of 64 units each and a standard wide policy network with two hidden layers of 400 and 300 units, respectively. In all cases, we also consider a variant of PPO that uses our adaptive learning rate method to better understand the drivers of performance in GePPO.

Figure 4 and Figure 5 show the performance throughout training and change in average total variation distance per policy update, respectively, when using the default policy network. Note that the results shown in Figure 1 and Figure 2 for PPO and GePPO with the default policy network are repeated here for reference. As discussed in Section 8, we see that the change in total variation distance per policy update in PPO increases throughout training across all environments. Because hyperparameters were tuned using the default policy network, this trend does not lead to unstable performance. The addition of our adaptive learning rate to PPO results in comparable performance in this well-tuned setting, while ensuring that the change in total variation distance remains close to the target determined by the clipping parameter. This causes a decrease in total variation distance change per policy update compared to PPO at the end of training in most environments, and an increase compared to PPO at the beginning of training in Swimmer-v3 and Hopper-v3. Finally, note that GePPO outperforms the variant of PPO with our adaptive learning rate, which indicates that sample reuse is an important driver of performance in GePPO. On average, GePPO improves average performance throughout training by 31% compared to PPO and 32% compared to PPO with our adaptive learning rate.

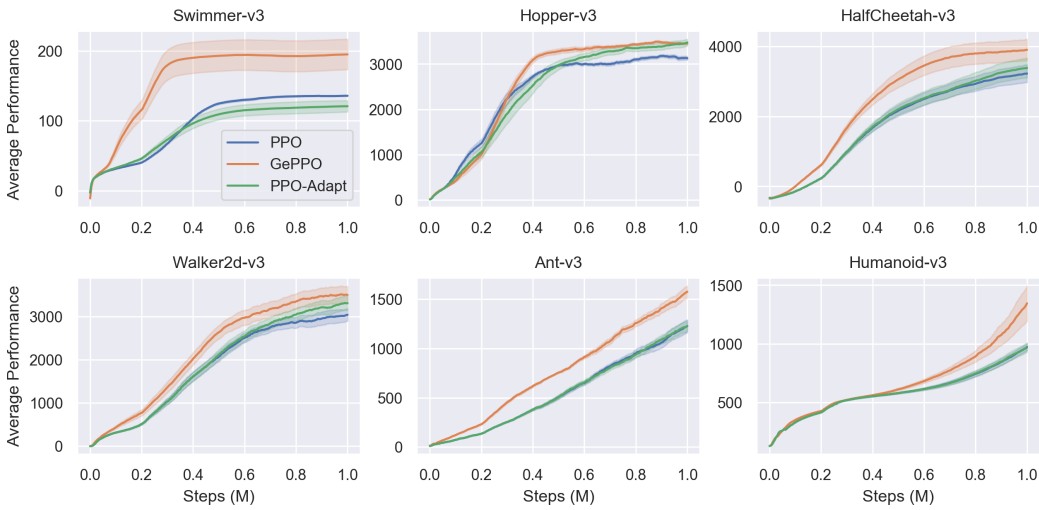

Figure 4: Performance throughout training across MuJoCo tasks, where the policy network has two hidden layers of 64 units each. Shading denotes half of one standard error. PPO-Adapt represents PPO with the addition of our adaptive learning rate method.

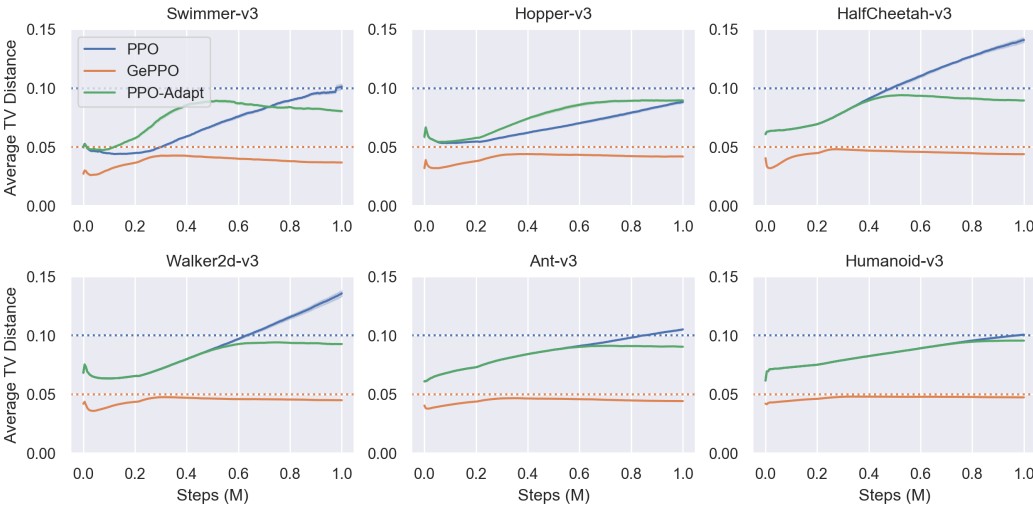

Figure 5: Change in average total variation distance per policy update across MuJoCo tasks, where the policy network has two hidden layers of 64 units each. Shading denotes half of one standard error. PPO-Adapt represents PPO with the addition of our adaptive learning rate method. Horizontal dotted lines represent target total variation distances for PPO and GePPO, respectively.

Figure 6 and Figure 7 show the same metrics for a standard wide policy network with two hidden layers of 400 and 300 units, respectively. The results shown in Figure 2 for Walker2d-v3 are repeated here for reference. As described in Section 8 for Walker2d-v3, we see that the wide policy network exacerbates the total variation distance trend observed in PPO under the default settings. The increased size of the policy network causes the probability ratio to be more sensitive to gradient updates during training, which renders the clipping mechanism ineffective and leads to excessively large policy updates. These large updates result in poor performance across several environments, and even cause performance to decline over time for Hopper-v3 and Walker2d-v3. The addition of our adaptive learning rate to PPO controls this source of instability, resulting in improved performance throughout training and stable policy updates with acceptable levels of risk. The exception to this trend is Humanoid-v3, where PPO outperforms PPO with our adaptive learning rate. This suggests that a more aggressive risk profile may improve performance on Humanoid-v3 without sacrificing

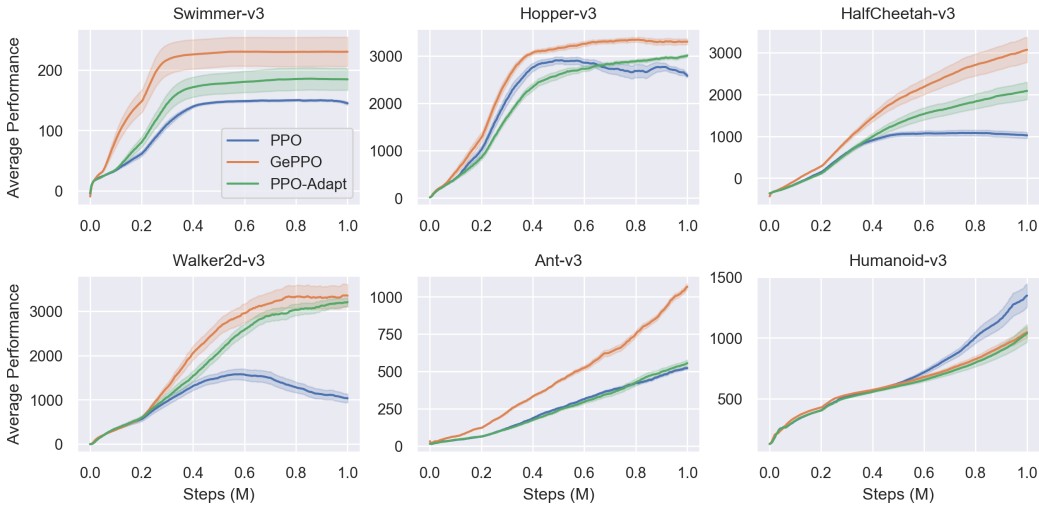

Figure 6: Performance throughout training across MuJoCo tasks, where the policy network has two hidden layers of 400 and 300 units, respectively. Shading denotes half of one standard error. PPO-Adapt represents PPO with the addition of our adaptive learning rate method.

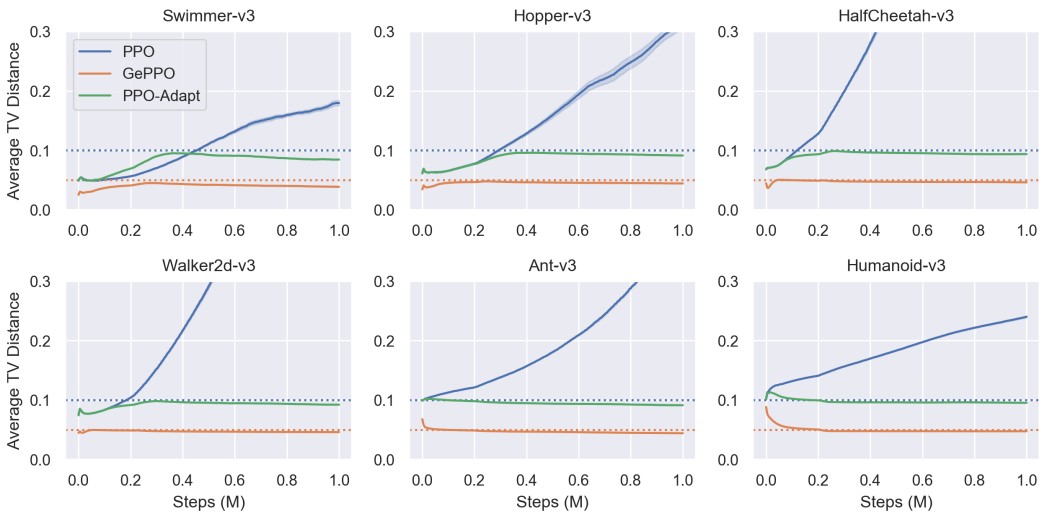

Figure 7: Change in average total variation distance per policy update across MuJoCo tasks, where the policy network has two hidden layers of 400 and 300 units, respectively. Shading denotes half of one standard error. PPO-Adapt represents PPO with the addition of our adaptive learning rate method. Horizontal dotted lines represent target total variation distances for PPO and GePPO, respectively.

training stability. Finally, by reusing samples from prior policies, GePPO further improves upon the performance of PPO with our adaptive learning rate. On average, GePPO improves average performance throughout training by 63% compared to PPO and 34% compared to PPO with our adaptive learning rate.

From these results, we see that the principled sample reuse in GePPO is a key driver of performance gains compared to PPO, while the use of an adaptive learning rate is important for stable performance that is robust to hyperparameter settings. Note that it is possible to achieve strong performance without the use of our adaptive learning rate, but this requires careful tuning of the learning rate that depends on several factors including the environment, policy network structure, length of training, and other hyperparameter settings.