# OpenReview forum: "Generalized Proximal Policy Optimization with Sample Reuse"
_NeurIPS.cc/2021/Conference — NeurIPS 2021 Poster_

### Official Review · Reviewer_CJcP · 2021-07-16

**Rating:** 6
**Confidence:** 5

**Summary:**


This paper extends PPO to an off-policy case. They develop policy improvement bound for the off-policy setting.


**Main Review:**


Generally, I like the idea of the paper. Extending PPO to off-policy case is an interesting and promising aspect. The paper has provided concise theorem derivation and an elegant realization of off-policy PPO. I think it's important and valuable.

I have gone through the detail of the derivation and the theorems seem correct for me. I like the insightful derivation in Theorem 1, Lemma 3, and Definition 1.

However, I do not give a high rating as the paper does not demonstrate the effectiveness of the methods (see the detail of the comment below).







The main weakness of the paper is the experiment.
- There are abundant environments in Mujoco and Atari. Since the author has provided a very general algorithm, I expect the author to conduct experiments on more environments to verify the effectiveness of the newly proposed methods.
- The comparison methods is missing.
A) Comparison with other PPO variants. There are several methods (as you have stated in related work) which can improve PPO while require equal computation as PPO. I expect the author to compare with these methods or extend their methods to these settings (since the method in the paper is very general and can also be extended to these PPO variants. I should not blame the author for this, but the author should verify the effectiveness of the proposed method).
B) Comparison with other off-policy methods, such as DDPG, SAC. One shortcoming of PPO is its on-policy limitation. I expect the proposed off-policy PPO could be competitive with other off-policy methods.
- Alation study on the parameter \epsilon. By the way, I don't understand why \epsilon is not presented in Algorithm 1 (Input) and APPENDIX Table 2.




Minor questions:
In my view, one critical component for success is the estimation of A^{\pi_k}(s,a) using the off-policy data of \pi_{k-i}. They use the V-trace method for estimation. I expect the author to discuss on this (not necessary).



**Time Spent Reviewing:**

9

---

> ### Author Response · Authors · 2021-08-09
> **Authors' Response**
>
> Thank you for your helpful comments and suggestions. We are glad you liked the idea of our paper, and we are excited to see that you believe our off-policy variant of PPO is an important and valuable contribution. As we detail below, we have doubled the number of MuJoCo locomotion tasks considered in the experiments section since submission, which we believe significantly strengthens our results. We hope that these additional results, along with our responses to your other comments, address your concerns regarding the experiments section.
>
> More environments:
>
> Since submission, we have added results on three additional locomotion tasks: Swimmer-v3, Ant-v3, and Humanoid-v3. Swimmer-v3 is a lower-dimensional task than the tasks considered in our submission, and Ant-v3 and Humanoid-v3 are significantly higher-dimensional tasks. GePPO demonstrates improved performance over PPO on these additional tasks, in line with the tasks considered in our original submission. This further demonstrates the stable and adaptive nature of GePPO, as it performs well across a range of task sizes. While our methods can be applied to both discrete and continuous action spaces, we chose to focus our experimental results on high-dimensional continuous control tasks which are arguably more difficult.
>
> Comparison with PPO variants:
>
> We have focused on comparing GePPO to the baseline PPO method in order to isolate the impact of our contributions compared to the most widely used version of PPO. However, we agree that the generality of our results allows our contributions to be combined with other improvements to PPO, and such an analysis represents an interesting extension for future work.
>
> Comparison with off-policy methods:
>
> We did not include popular off-policy algorithms as additional baselines in our experimental results since the risk profiles of these algorithms prevent a fair comparison. Our focus was on developing methods for principled sample reuse that retain the attractive property of approximate policy improvement guarantees, while existing off-policy algorithms such as DDPG and SAC do not satisfy these guarantees. Instead, they tend to be much more aggressive in their use of off-policy data in policy updates without directly controlling for the bias introduced. This can result in strong performance in certain environments such as HalfCheetah and Humanoid (see [12]), but also leads to a much higher level of potential risk and instability that is only controlled through careful, task-dependent hyperparameter tuning which is undesirable.
>
> Ablation of $\epsilon$:
>
> We have used the default choice of $\epsilon^{\mathrm{PPO}}=0.2$ for PPO, which is standard across the literature. This hyperparameter value appears in Table 2, but we will also add an explicit reference to this choice in the main experiments section.
>
> Given $\epsilon^{\mathrm{PPO}}$, GePPO automatically sets $\epsilon^{\mathrm{GePPO}}$ based on Lemma 4 to guarantee that PPO and GePPO are comparable in terms of their worst-case expected performance loss at every update. We will make this clear in the experiments section. As a result, for any change to $\epsilon^{\mathrm{PPO}}$ used in the baseline PPO method, $\epsilon^{\mathrm{GePPO}}$ used in GePPO would adjust accordingly to guarantee a fair comparison. It is also worth noting that our adaptive learning rate method allows us to avoid retuning the learning rate for different choices of $\epsilon^{\mathrm{PPO}}$. Based on your suggestion, we will consider other choices of $\epsilon^{\mathrm{PPO}}$ to include in a revised version of the supplementary material.
>
> Advantage estimation:
>
> Additional implementation details related to the use of V-trace for advantage estimation can be found in Section C of the Appendix. V-trace is a commonly used method for off-policy value function estimation and performs well in practice.

---

> > ### Comment · Reviewer_CJcP · 2021-08-21
> > **Response to Author**
> >
> > Thanks for your response, and it addressed some concerns of mine.
> > I'm glad that you have done experiments on the Humanoid and Ant task. These tasks are high-dimensional tasks, and off-policy methods can often be unstable. As you have said, "GePPO demonstrates improved performance over PPO on these additional tasks, in line with the tasks considered in our original submission". I would like you to provide the detailed score and the number of the seeds.

---

> > > ### Author Response · Authors · 2021-08-22
> > > **Detail on Additional Experimental Results**
> > >
> > > We are glad that we have been able to address some of your concerns. The additional experiments on Swimmer-v3, Ant-v3, and Humanoid-v3 follow the same experimental design used in the paper. We run each experiment for a total of one million steps over five random seeds. Because the dimensionality of Ant-v3 and Humanoid-v3 is much higher than the other tasks considered, we tuned the learning rate and initial standard deviation of the policy used for PPO on these tasks. For a fair comparison, we used the same learning rate and initial standard deviation of the policy for GePPO.
> > >
> > > See below for a summary of the results on Swimmer-v3, Ant-v3, and Humanoid-v3. You can also view revised versions of Table 1 and Figure 1 that include these additional results here: https://figshare.com/s/952bd83ec49d15442eb7
> > >
> > > Swimmer-v3:
> > >
> > > * Average Performance Over 1M Steps: PPO: 98; GePPO: 161 (+65%)
> > > * Final Performance: PPO: 136; GePPO: 195 (+44%)
> > > * Steps (M) to Final PPO Performance: PPO: 1.00; GePPO: 0.23
> > >
> > > Ant-v3:
> > >
> > > * Average Performance Over 1M Steps: PPO: 545; GePPO: 762 (+40%)
> > > * Final Performance: PPO: 1,227; GePPO: 1,576 (+28%)
> > > * Steps (M) to Final PPO Performance: PPO: 1.00; GePPO: 0.79
> > >
> > > Humanoid-v3:
> > >
> > > * Average Performance Over 1M Steps: PPO: 584; GePPO: 665 (+14%)
> > > * Final Performance: PPO: 972; GePPO: 1,345 (+38%)
> > > * Steps (M) to Final PPO Performance: PPO: 1.00; GePPO: 0.85

---

> > > > ### Comment · Reviewer_CJcP · 2021-09-02
> > > > **Response to authors**
> > > >
> > > > I thank the efforts of the authors on the results of new experiments. I have also read the response to other reviewers, and the explanations is satisfactory to me. I will keep my original score.

---

### Official Review · Reviewer_2GSR · 2021-07-16

**Rating:** 7
**Confidence:** 4

**Summary:**

This work generalizes PPO to the off-policy setting. First the policy improvement bound from Achiam et al. (2017) was generalized to accomodate for several prior policies. The authors also made a connection between the total variation distance and the PPO clipping term. A new algorithm GePPO is proposed which is shown to be more sample efficient compared to standard PPO.

**Limitations And Societal Impact:**

See main review

**Main Review:**

Overall I think this is a very well-written paper. To the best of my knowledge, this work is the first to draw a direct connection between the monotonic policy improvement theorem and the clipped PPO algorithm, including deriving a relationship between the clipping mechanism and the total variational distance. The generalization of the policy improvement bound to past policies is also quite innovative. Most of the proofs in this paper directly extend upon the proofs in Achiam et al. (2017) and do appear to be correct.  Different components of the new algorithm were thoroughly analyzed. Experiments presented in this paper are mostly quite well designed though I think a comparison with some popular off-policy algorithms and also TRPO (which is a more direct application of the policy improvement bound) would be of interest. I also particularly like the fact that the authors are very forthcoming with the limitations of their paper, since the proposed algorithm is a variation of PPO, I do not expect it to achieve SOTA performance but I believe this work does provide additional useful insights into the PPO algorithm and serves as a useful jumping off point for the community for further research.


**Time Spent Reviewing:**

4-5

---

> ### Author Response · Authors · 2021-08-09
> **Authors' Response**
>
> Thank you for your thoughtful feedback. We appreciate your comments on the novelty of our generalized lower bound and the value of our connections between this bound and PPO. We hope that our work provides useful insights for further research in the field as you have mentioned.
>
> We are happy to share that we have further strengthened our experimental evidence by adding results on additional locomotion tasks: Swimmer-v3, Ant-v3, and Humanoid-v3. Swimmer-v3 is a lower-dimensional task than the tasks considered in our submission, and Ant-v3 and Humanoid-v3 are significantly higher-dimensional tasks. GePPO demonstrates improved performance over PPO on these additional tasks, in line with the tasks considered in our original submission. This further demonstrates the stable and adaptive nature of GePPO, as it performs well across a range of task sizes.
>
> Next, we comment on the comparisons of interest mentioned in your review:
>
> * Comparison with TRPO: We considered PPO in this paper since it is the most widely used on-policy algorithm, and chose to focus on analyzing the improvements of our approach over the baseline PPO; thus, we did not include TRPO in our experimental results. However, we wish to note that our generalized policy improvement lower bound in Theorem 1 can also be used to motivate an off-policy variant of TRPO using techniques similar to those used to develop GePPO. We leave this interesting extension for future work, but we expect such a generalized version of TRPO would lead to similar improvements over the baseline TRPO as observed for GePPO vs. PPO.
>
> * Comparison with off-policy methods: We did not include popular off-policy algorithms as additional baselines in our experimental results since the risk profiles of these algorithms prevent a fair comparison. Our focus was on developing methods for principled sample reuse that retain the attractive property of approximate policy improvement guarantees, while existing off-policy algorithms do not satisfy these guarantees. Instead, they tend to be much more aggressive in their use of off-policy data in policy updates without directly controlling for the bias introduced. This can result in strong performance in certain environments such as HalfCheetah and Humanoid (see [12]), but also leads to a much higher level of potential risk and instability that is only controlled through careful, task-dependent hyperparameter tuning which is undesirable.

---

> > ### Comment · Reviewer_2GSR · 2021-09-01
> > **Post-rebuttal response**
> >
> > I want to thank the authors for their detailed response, I am satisfied with how my concerns were addressed. The new experiments in responding to other reviewers comments are also much appreciated. I am therefore keeping my original score.

---

### Official Review · Reviewer_FiaV · 2021-07-16

**Rating:** 7
**Confidence:** 3

**Summary:**

The paper proposes a generalization of PPO to the off-policy setting.

PPO has been extensively used with multiple policy updates, making it inherently off-policy and it was no clear how the algorithm should be modified in order to preserve its theoretical guarantees.

**Limitations And Societal Impact:**

Yes

**Main Review:**

The adaptation or the bound from Achiam follows easily when considering a distribution over the last M policies. The additional theoretical results help understanding how one should choose this distribution or adapt the learning rate of the policy in a more principled way. The authors perform an ablation, showing that the benefits in performance come both from the adapted clipping mechanism and from the reuse of past data.

I think it would be interesting to compare this also to the case when the generalized clipping mechanism is missing


**Time Spent Reviewing:**

5

---

> ### Author Response · Authors · 2021-08-09
> **Authors' Response**
>
> Thank you for taking the time to review our paper. We appreciate the positive feedback on our theoretical results and how they motivate a principled approach to sample reuse in PPO.
>
> We are happy to share that we have further strengthened our experimental evidence by adding results on additional locomotion tasks: Swimmer-v3, Ant-v3, and Humanoid-v3. Swimmer-v3 is a lower-dimensional task than the tasks considered in our submission, and Ant-v3 and Humanoid-v3 are significantly higher-dimensional tasks. GePPO demonstrates improved performance over PPO on these additional tasks, in line with the tasks considered in our original submission. This further demonstrates the stable and adaptive nature of GePPO, as it performs well across a range of task sizes.
>
> We are somewhat unclear on what you mean when you refer to the case when the generalized clipping mechanism is missing. The generalized clipping mechanism in Definition 1 provides a method for controlling the penalty term in the generalized policy improvement lower bound in Theorem 1, allowing for the principled use of off-policy data in PPO. We will comment on a couple of possibilities:
>
> * The results for PPO-Adapt in Section D of the Appendix represent standard on-policy PPO with the addition of our adaptive learning rate (but without sample reuse with our generalized clipping mechanism). These results demonstrate that the adaptive learning rate leads to stable performance that is robust to hyperparameter settings, but the addition of sample reuse with our generalized clipping mechanism further improves average performance throughout training over PPO-Adapt.
>
> * It is also possible that you are referring to a version of GePPO without an adaptive learning rate. Similar results for GePPO can be achieved without an adaptive learning rate by selecting an appropriate fixed learning rate, but our methodology allows us to avoid the task-dependent hyperparameter tuning required to determine the appropriate fixed learning rate.

---

> > ### Comment · Reviewer_FiaV · 2021-08-26
> > **Response to authors**
> >
> > I thank the authors for the detailed answer. The authors clarified my doubts and included additional results showing that their method is stable even when using higher-dimensional environments. Overall I think this paper will be a nice contribution to the community and I recommend accepting it.

---

### Official Review · Reviewer_nKjh · 2021-07-20

**Rating:** 6
**Confidence:** 3

**Summary:**

This paper presents a novel policy improvement lower bound that provides theoretical evidence for the off-policy sample reuse of PPO algorithms. Specifically, the theorem justifies the clip
term in PPO as a regularization in the lower bound.

**Limitations And Societal Impact:**

There is no potential negative societal impact.

**Main Review:**

### Methodology

The proposed learning rate schedule is similar to the adaptive Adaptive KL penalty coefficient in PPO. Could the author comment on which schema has better control on the TV?


Presumably, the older policy will have larger TV divergence comparing to the more updated policies,
is there a better approach than uniform weight? Does this weighting strategy affect a lot?

### Experiments

The experimental results of this work are relatively weak. 1) The common practice in the literature of sample-efficiency algorithms [continuous action space] benchmark six mujoco environments. 2) The only baseline in this work is PPO. It would be helpful to also compare with off-policy algorithms [TD3, SAC, DDPG] as this work enables the sample reuse.

When comparing to the baselines, I think the fair comparison would be set the replay ratio same as the baseline. Currently, the replay ratio is B times of PPO. For off-policy approaches, the replay ratio can affect performance a lot.

Furthermore, PPO is not restricted to continuous action environments. It would be more solid to also benchmark the proposed algorithm on discrete action space environments [e.g., Atari].

It would be interesting to show the balance between wall clock time and sample efficiency. How does the proposed regularization affect the wall clock time?

Figure 2 indicates the performance drop is aligned with the change in total variation distance is inspiring.

It looks like most of the performance gain is from the adaptation of learning rate based on the results of
PPO-Adapt in the appendix. It would be helpful to see what is the performance of GePPO without learning rate adaptation.





**Time Spent Reviewing:**

3

---

> ### Author Response · Authors · 2021-08-09
> **Authors' Response**
>
> Thank you for taking the time to review our paper. See below for responses to your questions, which include references to where information can be found in the paper, where appropriate. Importantly, we have doubled the number of MuJoCo locomotion tasks considered in the experiments section (from 3 to 6), which we believe significantly strengthens our results. We hope that our improved experimental evidence, along with our responses to your other comments, address your concerns.
>
> Experiments:
>
> * Additional environments: Since submission, we have added results on three additional locomotion tasks: Swimmer-v3, Ant-v3, and Humanoid-v3. Swimmer-v3 is a lower-dimensional task than the tasks considered in our submission, and Ant-v3 and Humanoid-v3 are significantly higher-dimensional tasks. GePPO demonstrates improved performance over PPO on these additional tasks, in line with the tasks considered in our original submission. This further demonstrates the stable and adaptive nature of GePPO, as it performs well across a range of task sizes. While our methods can be applied to both discrete and continuous action spaces, we chose to focus our experimental results on high-dimensional continuous control tasks which are arguably more difficult.
>
> * Comparison with off-policy methods: We did not include popular off-policy algorithms as additional baselines in our experimental results since the risk profiles of these algorithms prevent a fair comparison. Our focus was on developing methods for principled sample reuse that retain the attractive property of approximate policy improvement guarantees, while existing off-policy algorithms do not satisfy these guarantees. Instead, they tend to be much more aggressive in their use of off-policy data in policy updates without directly controlling for the bias introduced. This can result in strong performance in certain environments such as HalfCheetah and Humanoid (see [12]), but also leads to a much higher level of potential risk and instability that is only controlled through careful, task-dependent hyperparameter tuning which is undesirable.
>
> * Interpretation of results: We believe that the experimental results demonstrate the importance of both the adaptive learning rate and of sample reuse. We provide a detailed discussion supporting this claim in Section D of the Appendix. Similar results for GePPO can be achieved without an adaptive learning rate by selecting an appropriate fixed learning rate, but our methodology allows us to avoid the task-dependent hyperparameter tuning required to determine the appropriate fixed learning rate.
>
> * Wall-clock time: We provide commentary on wall-clock time in Section C of the Appendix under Computational Resources. The wall-clock time of GePPO is slightly higher than PPO due to the fact that GePPO performs twice as many updates in our experiments compared to PPO. Although the focus of our algorithm is on sample efficiency, GePPO also leads to improved performance compared to PPO for a fixed amount of wall-clock time.
>
> Clarifications on methodology:
>
> * Adaptive learning rate vs. KL penalty: The probability ratio clipping objective in Equation (2) has become the default implementation of PPO due to its superior performance, rather than the version with an adaptive KL penalty term that you have mentioned. We have demonstrated how this clipping objective is closely related to controlling the TV distance penalty term in the policy improvement lower bound, and our adaptive learning rate allows this to hold in practice. This connection is one possible reason for why the clipping objective works better than the adaptive KL penalty term, since the KL penalty term is motivated by bounding the TV distance penalty term via Pinsker’s inequality.
>
> * Policy weights: In Section 7, we present our sample efficiency analysis using uniform policy weights for concreteness. However, in Section B of the Appendix we demonstrate how these policy weights can be optimized. All experimental results use optimized policy weights based on Theorem 5. These weights are shown in the right-hand side of Figure 3.
>
> * Replay ratio: We consider n=1,024 samples as the minimum batch size for data collection (which represents as little as ~1 trajectory), and we perform policy updates in GePPO after every n samples. The number of prior policies M used for policy updates and the size of each policy update in GePPO are theoretically supported, and are chosen to guarantee improved stability and sample efficiency compared to PPO while maintaining the same level of risk per policy update. These inputs are difficult to compare to choices in popular off-policy algorithms such as replay ratio, replay buffer size, and number of gradient steps, which are treated as hyperparameters and are not chosen to control the bias introduced by off-policy data in a principled way.

---

> > ### Comment · Reviewer_nKjh · 2021-08-22
> > **reply to authors**
> >
> > Thanks for the detailed response.
> >
> > The author mentioned additional experiments on Mujoco showing consistent improved results over the PPO. I would like to believe the author they get the results but it would better to see the table of results e.g., the performance comparison at 1M step.
> >
> > It is not very convincing to me that comparing GePPO with other off-policy approach is not fair. The proposed approach working on the same off-policy setting while other SOTAs are not compared.
> >
> > I think I will maintain my evaluation for now.

---

> > > ### Author Response · Authors · 2021-08-24
> > > **Detail on Experimental Results**
> > >
> > > For ease of reference, see below for a summary of our additional results on Swimmer-v3, Ant-v3, and Humanoid-v3. We have also updated Table 1 and Figure 1 to include these results, which you can view here: https://figshare.com/s/952bd83ec49d15442eb7
> > >
> > > Swimmer-v3:
> > >
> > > * Average Performance Over 1M Steps: PPO: 98; GePPO: 161 (+65%)
> > > * Final Performance: PPO: 136; GePPO: 195 (+44%)
> > > * Steps (M) to Final PPO Performance: PPO: 1.00; GePPO: 0.23
> > >
> > > Ant-v3:
> > >
> > > * Average Performance Over 1M Steps: PPO: 545; GePPO: 762 (+40%)
> > > * Final Performance: PPO: 1,227; GePPO: 1,576 (+28%)
> > > * Steps (M) to Final PPO Performance: PPO: 1.00; GePPO: 0.79
> > >
> > > Humanoid-v3:
> > >
> > > * Average Performance Over 1M Steps: PPO: 584; GePPO: 665 (+14%)
> > > * Final Performance: PPO: 972; GePPO: 1,345 (+38%)
> > > * Steps (M) to Final PPO Performance: PPO: 1.00; GePPO: 0.85
> > >
> > > GePPO can be best characterized as an on-policy algorithm that incorporates samples from recent policies while maintaining approximate policy improvement guarantees. As shown by the theoretical analysis in Section 7, reusing data from recent policies improves the tradeoff between stability and sample efficiency in PPO. This is the focus of our paper, and our experiments provide empirical validation that our theoretical results hold in practice.
> > >
> > > Popular off-policy algorithms such as DDPG, TD3, and SAC, on the other hand, are not motivated by policy improvement guarantees. They reuse samples from policies that are significantly older than those considered in GePPO, which results in a very different risk profile compared to GePPO that prevents a fair comparison.
> > >
> > > While both classes of algorithms are useful, algorithms with performance guarantees are important for many real-world decision making settings. Policy improvement guarantees allow practitioners to trust that an algorithm will deliver safe and reliable performance throughout training, which is critical in high-stakes tasks such as autonomous driving and healthcare. Our work improves this important class of reinforcement learning algorithms through principled sample reuse, which we believe is a valuable contribution.

---

> > > > ### Comment · Reviewer_nKjh · 2021-08-25
> > > > **Response to the authors**
> > > >
> > > > Thanks for the detailed results and explanation.
> > > >
> > > > The experimental results looks ok to me for now. I increased my rating for the verified improvements comparing to PPO.
> > > >
> > > > It is not very clear to me if it is appropriate to claim this is an off-policy version as this term refers to using arbitrary data collected from any behavior policies for model updates.

---

### Decision · Program_Chairs · 2021-09-27

**Decision:**

Accept (Poster)

**Comment:**

After reading each other's reviews and the authors' feedback, the reviewers discussed the merits and flaws of the paper.
The authors' answers solved most of the doubts exposed in the reviews, and the reviewers agree that this paper can be accepted for publication.
I want to congratulate the authors and invite them to modify their paper following the reviewers' suggestions.